# POST-HOC SALIENCY METHODS FAIL TO CAPTURE LATENT FEATURE IMPORTANCE IN TIME SERIES DATA

**Maresa Schröder**[1,2*]**, Alireza Zamanian**[3,1*]**& Narges Ahmidi**[1]

[1] Fraunhofer Institute for Cognitive Systems IKS, Germany
[2] Technical University of Munich, Germany; TUM School of Computation, Information and Technology, Department of Mathematics
[3] Technical University of Munich, Germany; TUM School of Computation, Information and Technology, Department of Computer Science
{maresa.schroeder,alireza.zamanian,narges.ahmidi}@iks.fraunhofer.de

## ABSTRACT

Saliency methods provide visual explainability for deep image processing models by highlighting informative regions in the input images based on feature-wise (pixels) importance scores. These methods have been adopted to the time series domain, aiming to highlight important temporal regions in a sequence. This paper identifies, for the first time, the systematic failure of such methods in the time series domain when underlying patterns (e.g., dominant frequency or trend) are based on latent information rather than temporal regions. The latent feature importance postulation is highly relevant for the medical domain as many medical signals, such as EEG signals or sensor data for gate analysis, are commonly assumed to be related to the frequency domain. To the best of our knowledge, no existing post-hoc explainability method can highlight influential latent information for a classification problem. Hence, in this paper, we frame and analyze the problem of latent feature saliency detection. We first assess the explainability quality of multiple state-of-the-art saliency methods (Integrated Gradients, DeepLift, Kernel SHAP, Lime) on top of various classification methods (LSTM, CNN, LSTM and CNN trained via saliency-guided training) using simulated time series data with underlying temporal or latent space patterns. In conclusion, we identify that Integrated Gradients and DeepLift, if redesigned, could be potential candidates for latent saliency scores.

## 1 INTRODUCTION

Saliency methods aim to explain the predictions of deep learning models by highlighting important input features. These methods often assign scores to individual inputs (Guidotti et al., 2018; Ismail et al., 2020), collectively resulting in the detection of class-distinctive patterns. For image data, this means assigning scores to positional information, namely pixels. Such a strategy suits image data, as the label is often associated with specific input regions. Recently, image saliency methods have been adopted for time series data Loeffler et al. (2022); Schlegel et al. (2020). They similarly assign importance scores to the pixel counterparts, namely "time points". These methods suit the time series problem when a temporal pattern is indicative of the class. In some time series problems, however, the label may depend on latent features such as dominant frequency, state-space model parameters, or the overall trend of a non-stationary time series. In these cases, even though the classifier might successfully capture the latent space, the positional scores extracted from the classifier will not directly explain the importance of the underlying latent features. Hence, the generated saliency maps will not be directly interpretable and thus fail to fulfill their purpose.

The goal of this paper is to introduce, formulate and analyze the problem of latent feature saliency in deep time series classification problems, focusing on the fundamental Fourier series latent model. By extension, our study is replicable for other latent models. We summarize our main contributions below:

---

*Authors contributed equally

1. We draw attention to the problem of latent feature saliency detection in time series data. We formulate the shapelet- vs. latent-based pattern in time series classification and propose a definition for an ideal latent feature saliency method (Section 2).

2. We provide a comprehensive study of popular time series saliency methods, including Integrated Gradients, DeepLift, Kernel SHAP and Lime (Section3,Section 4) on top of multiple classification methods (LSTM, CNN, LSTM and CNN trained via saliency guided training).

3. We identify effective methods that can be extended to potentially tackle the problem of latent space saliency (Section 5).

## 2 PROBLEM FORMULATION

Let $D = (X, Y)$ with a univariate time series $X \in \mathcal{X}$ and the binary label $Y \in \{0, 1\}$ formulate a time series classification data set. Furthermore, let the mapping $f_{XY} : \mathcal{X} \mapsto \{0, 1\}$ represent a deep learning-based classifier. In *latent-representation learning*, we assume a latent space $\mathcal{Z}$, a mapping from feature to latent space $f_{XZ} : \mathcal{X} \mapsto \mathcal{Z}$ and a latent space to label mapping $f_{ZY} : \mathcal{Z} \mapsto \{0, 1\}$, such that the classifier $f_{XY}$ can be learned via the feature-to-latent and latent-to-label mappings.This view has been adopted by several time series classifiers such as hidden Markov models (HMM) and recurrent neural networks (RNN). The learned latent representation, exhibits properties shown to be significant in terms of explainability Mikolov et al. (2013); Charte et al. (2020). Instead of estimating $f_{XZ}$ as a black-box model, a parametric latent model (such as Fourier series models, state space models, linear and switching dynamical systems, or additive and multiplicative models) can be estimated via a neural network. These models are motivated by prior knowledge about the underlying data generation mechanism; thus, their parameters often are interpretable. A saliency method applied to this solution assigns scores to latent features in the $\mathcal{Z}$ space. In contrast, methods used for the black-box models usually lack explainability for the latent features.

The latent space assumption is relevant in many time series problems. Sound signals are often differentiated by amplitude and frequency; thus, the decision process behind audio classification is likely to be better explained by the Fourier latent space than by spatial importance scores. Vibration signal classification, as in earthquake or production line failure prediction, is likely to also depend on frequency or amplitude. Financial time series classification often revolves around modeling trends and seasonality of the time series. Many signals in the medical domain, such as EEG or sensor data from wearable technologies for gait analysis for neurological disease progression, pain recognition, or medication level adjustment, are further strongly related to amplitude and frequency. These examples show that achieving time series explainability is heavily related to latent space assumptions.

### 2.1 LATENT FEATURES VS. SHAPELETS

Ye & Keogh (2009) define *shapelets* as variable-length subsequences of time series which are maximally representative of a class. We define a feature-to-shapelet mapping $f_{XS} : \mathcal{X} \mapsto [0, 1]^k$. Samples in $\mathcal{S}$ are normalized score vectors, determining which shapelet appears in a sample. Subsequently, shapelet-based classifiers predict the label based on an existing pattern in the time domain. These models are coordinated with saliency methods, which in this case, are visually explainable since time points are directly expressive of both saliency scores and shapelets. The presence of informative shapelets does not contradict the assumption of a latent model. On the contrary, shapelets may appear as a proxy for latent information (see Figure 2). Nevertheless, from the explainability point of view, there is a notable difference between latent features and shapelets. As an example, a label correlated with the damping ratio of a vibration signal can be potentially predicted by shapelet-based classifiers; however, a conventional saliency method applied to this problem will only highlight a proxy of the informative latent feature, namely the existing fluctuations and oscillations of the time series. In conclusion, time series classification problems may be characterized by class differences in features that belong to the time domain as shapelets or to a latent domain. Current saliency methods can provide explainability for shapelets but not directly for latent models.

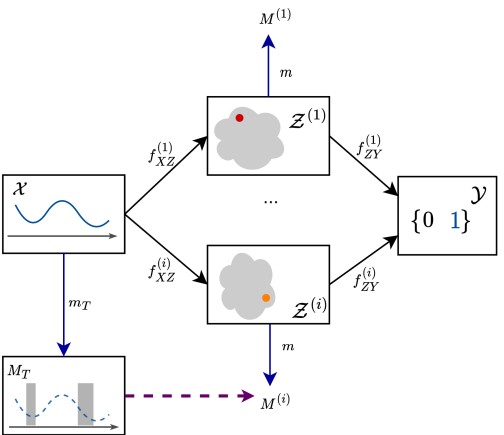

Figure 1: Time series classification schematic over the space $\mathcal{X} \times \mathcal{Y}$ with latent space representations $Z^{(i)}$, associated with saliency function $m(Z^{(i)})$ and resulting saliency map $M^{(i)}$. Current methods $m_T$ measure saliency of the feature space, yielding the map $M_T$.

## 2.2 Defining a desirable saliency method for time series

Figure 1 illustrates the setup of a time series classification problem with multiple possible intermediate latent spaces, enumerated with $i$, and denoted as $\mathcal{Z}^{(i)}$. A time series $X \in \mathcal{X}$ can be mapped to $\mathcal{Z}^{(i)}$ by the $i$-th chosen latent model $f_{XZ}^{(i)}$. Without loss of generality, we assume that there is only one latent feature $Z^*$ which provides the best explanation for the classification task. The latent space that contains $Z^*$ is denoted as $\mathcal{Z}^{(*)}$.

We define a saliency method as "reliable" if it assigns the highest score to $Z^*$ above all other features throughout all latent spaces. To formulate the reliability definition, we consider a *latent-aware* saliency method $m : \mathcal{Z}^{(i)} \mapsto \mathbb{R}_+^{|Z^{(i)}|}$, which produces a saliency map $M^{(i)}$ for $\mathcal{Z}^{(i)}$. The reliability condition is then formulated as

$$\forall i \neq *, \quad \max M^{(*)} > \max M^{(i)}.$$

Note that during implementation, we have to define the possible set of latent models manually.

The fundamental problem of existing saliency methods is that they only estimate the saliency map for the time domain and therefore lack appropriate output for features in other domains. Hence none of the existing saliency methods meet the criteria for reliability. However, we argue that there might exist some *promising failing methods*, which require only minor adjustments to serve as desired saliency methods for time series. We define a saliency method $m_T : \mathcal{X} \mapsto \mathbb{R}_+^{|X|}$ as promising if the produced map $M_T \in \mathbb{R}_+^T$ bears enough information to infer $M^{(i)}, \forall i$ (possibly via a simple mapping function, depicted as a purple arrow in Figure 1). In other words, $m_T$ can capture information about

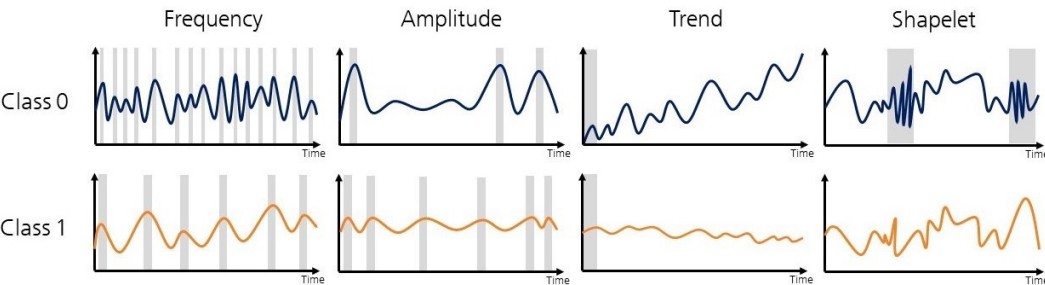

Figure 2: Toy examples of multiple label-making scenarios. Influential time steps (regions with high saliency scores) are shaded in grey for frequency (peaks), amplitude (highest peaks), trend (a window enough for inferring about the trend), and shapelet (presence of the informative pattern).

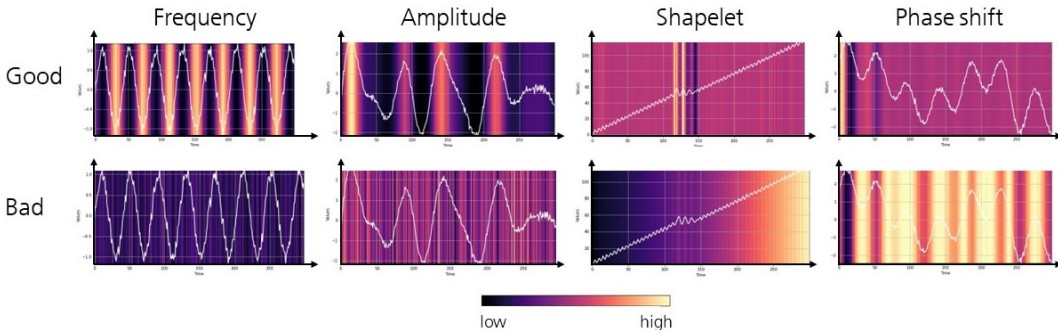

Figure 3: Examples of well-performing explainability methods (top row) providing to some extend interpretable explanations and completely uninterpretable saliency results (bottom row). Saliency scores are visualized as a background heat map.

latent saliency, even though it cannot directly explain it. In this case, an extension of the promising method, representing the mapping from $M_T$ to $M^{(*)}$, establishes a desired latent saliency method.

Figure 2 schematically depicts the output of a good failing method when the label is associated with either the frequency or amplitude of a Fourier model, the trend of an additive model, or shapelets. In particular, highlighted regions are sufficient to infer the latent parameter (or equally shapelet). Putting the experiment into practice, Figure 3 presents heat maps of importance scores resulting from two exemplary failing methods.

## 3 EXPERIMENTAL FRAMEWORK

As a preliminary step for presenting the results of the empirical study, this section introduces the examined time series saliency methods, data sets and the implementation details.

Our study focuses on *post-hoc* saliency methods designed to explain single classification instances of trained models. Here, we investigate the following state-of-the-art saliency methods and group them into three families.

(1) *Gradient-based feature attribution (FA)* methods infer input feature importance based on the magnitude of the gradient of the output with respect to the input features. The attribution method Saliency Simonyan et al. (2014) directly employs gradients to generate saliency maps. Extensions of this basic method are Gradient × Input Shrikumar et al. (2016), DeCovNet Zeiler & Fergus (2014), Guided Backpropagation Springenberg et al. (2015) and SmoothGrad Smilkov et al. (2017). Deep-Lift Shrikumar et al. (2017) utilizes a neuron attribution-based difference-from-reference approach to assigning scores. Integrated Gradients (IG) Sundararajan et al. (2017) calculates the path integral from a non-informative baseline input to the respective input feature, tackling the problem of gradient saturation Bastings & Filippova (2020). Relevance-based methods, e.g., Layer-wise Relevance Propagation (LRP) Bach et al. (2015) and Deep Taylor Decomposition Montavon et al. (2017), calculate attribution scores by propagating relevance scores from the output back through the network via designed propagation rules.

(2) *Model-agnostic* FA methods can be applied to any black-box classifier without access to the models' parameters Carrillo et al. (2021); Petsiuk et al. (2018). Methods such as Occlusion Zeiler & Fergus (2014), Meaningful Perturbations Fong & Vedaldi (2017) and RISE Petsiuk et al. (2018) assign saliency scores relative to the change in output when the respective feature is perturbed. LIME Ribeiro et al. (2016) fits local interpretable surrogate models to the classifier in the neighborhood of the target sample and calculates the saliency based on these models' parameters. Other methods are inspired by theorems from the field of game theory Datta et al. (2016); Lipovetsky & Conklin (2001); Štrumbelj & Kononenko (2014). In particular, the application of the Shapley Value Shapley (1953) has achieved great popularity. Lundberg & Lee (2017) introduce the SHAP values method to measure feature importance by the Shapley value of a conditional expectation function of the to-be-explained model.

(3) A different class of post-hoc methods generates *counterfactual explanations* (CF) as LASTS Guidotti et al. (2020), time series tweaking Karlsson et al. (2020), LatentCF++ Wang et al. (2021), CoMTE Ates et al. (2021) and Native Guide Delaney et al. (2021). These methods identify counter-samples to provide explainability by estimating the required variation in individual input features to change the classification outcome. Since our experiments focus on saliency maps, we exclude CF methods from our investigations in this paper.

For our study, we selected four candidate methods from different classes of post-hoc methods: Integrated Gradients (IG), Deep-Lift (DL), LIME and Kernel SHAP (SHAP). As for the classifiers, we utilize long-short term memory networks (LSTM) Hochreiter & Schmidhuber (1997) and convolutional neural networks (CNN) Le Cun et al. (1989). Since the experiments focus on saliency detection, we also train the LSTM and CNN networks via a saliency-guided training procedure (SGT) Ismail et al. (2021). This procedure allows networks to produce more consistent saliency scores, as the saliency feedback is used for training the network.

### 3.1 Data set generation

To demonstrate our findings, we designed a simulation study in which time-series data is generated based on the Fourier series model. The Fourier series is a well-known latent model for many natural scenarios Geweke & Singleton (1981); Bracewell (2000) and it is proven that any given univariate time series can be reconstructed from its Fourier latent space using a Fourier transformation function. The Fourier latent space can be defined as a matrix with three rows representing frequencies, amplitudes and phase shifts. In our experiments, the Fourier latent space is a matrix of 3x10 parameters.

We generated a total of ten experiments to understand the response of different saliency methods to different patterns. Our ten experiments include four experiments with temporal shapelet patterns, two with latent amplitude patterns, two with latent frequency patterns, and two with latent phase shift patterns. In each experiment, we build a data set containing 2560 time series samples of equal length divided into two equally sized classes. For the shapelet experiments, each sample in the data set is generated by first randomly sampling from the latent space and then applying a Fourier transformation to reconstruct its temporal signal from the latent space matrix. Afterward, the time series samples in class 1 were superimposed with a dominant shapelet pattern positioned either at a random location (experiment 1), the end (experiment 2), middle (experiment 3) or start (experiment 4) of the time series. For the latent feature experiments, the latent space matrices for class 0 were sampled from a latent space different than the latent space for class 1. The difference was defined in terms of sampling intervals for frequency, amplitude or phase shift. A detailed description of the sampling distributions per experiment is presented in Table 3 in Appendix A.2. For each experiment, the training, validation and testing sets were generated by random sampling without replacement with a ratio of 80%, 10% and 10%, respectively.

For assigning the labels to the data samples, we induced a simple linear relation between the latent or temporal patterns and the class labels. In the latent scenarios, two classes are distinguishable using a single decision boundary defined as $Z^* = const.$, meaning that only one latent feature is class-distinctive. Likewise, in shapelet-related scenarios, the presence or absence of a specific shapelet decides the label of the data. This allows us to study the latent features individually and in a controlled manner. In such settings, potential poor results can be confidently attributed to the intrinsic weakness of the saliency methods rather than inappropriate classifiers. The data generation mechanism and the resulting data sets are presented and described in detail in Appendix A.1 and A.2, respectively.

### 3.2 Implementation details

In this paper, we investigate the performance of both the classifiers and the saliency methods with a particular focus on the interpretability of saliency methods. To ensure uniform power between all classifiers, they were designed as simple one-layer networks with no dropouts or other forms of additional regularization. The performance of saliency methods is strongly correlated with the classification performance, which is typically increased through more sophisticated and deeper networks. Therefore, by keeping the architecture simple, we intended to objectively evaluate and com-

pare the explainability methods without the influence of optional variations, preventing overfitting or performance boosting.

All algorithms were implemented in the Python programming language. The classifiers were implemented using the deep learning library *PyTorch* Paszke et al. (2019) with the help of the wrapper *PyTorch Lightning* Falcon (2019). Hyper-parameter optimization was performed through the library *Optuna* Akiba et al. (2019). For the feature attribution techniques, the implementations from the PyTorch-based model interpretability library *Captum* Kokhlikyan et al. (2020) were employed.

## 4 RESULTS

Table 1 reports the average accuracy and F1 scores of the chosen classifiers across our ten data sets grouped by the type of experiments. The results show that overall the CNN trained via saliency-guided training achieves the highest classification performance.

Table 1: Average classification performance on test data across all synthetic data sets.

| | Shapelet | | Frequency | | Phase shift | | Amplitude | |
| Classifier | Accuracy | F1 | Accuracy | F1 | Accuracy | F1 | Accuracy | F1 |
| --- | --- | --- | --- | --- | --- | --- | --- | --- |
| LSTM | 0.8535 | 0.8466 | 0.9749 | 0.9470 | 0.5157 | 0.4914 | 0.9981 | 0.9981 |
| LSTM + SGT | 0.8242 | 0.8417 | 0.9082 | 0.9117 | 0.5352 | 0.4145 | 0.9160 | 0.9230 |
| CNN | 0.6221 | 0.7439 | **0.9610** | **0.9633** | 0.9629 | 0.9625 | 0.9981 | 0.9981 |
| CNN + SGT | **0.8721** | **0.9138** | **0.9610** | **0.9633** | **0.9649** | **0.9634** | **1.0000** | **1.0000** |

It appears that the LSTM classifier is seriously challenged during phase-shift experiments. This could be due to the *vanishing gradient* problem of LSTMs, which hinders proper classification if informative patterns are placed in the early time points. Surprisingly, the LSTM with the saliency-guided training procedure performs slightly worse than the LSTM. Unlike the LSTM, the CNN largely benefits from the saliency-guided training procedure, especially in the shapelet experiments.

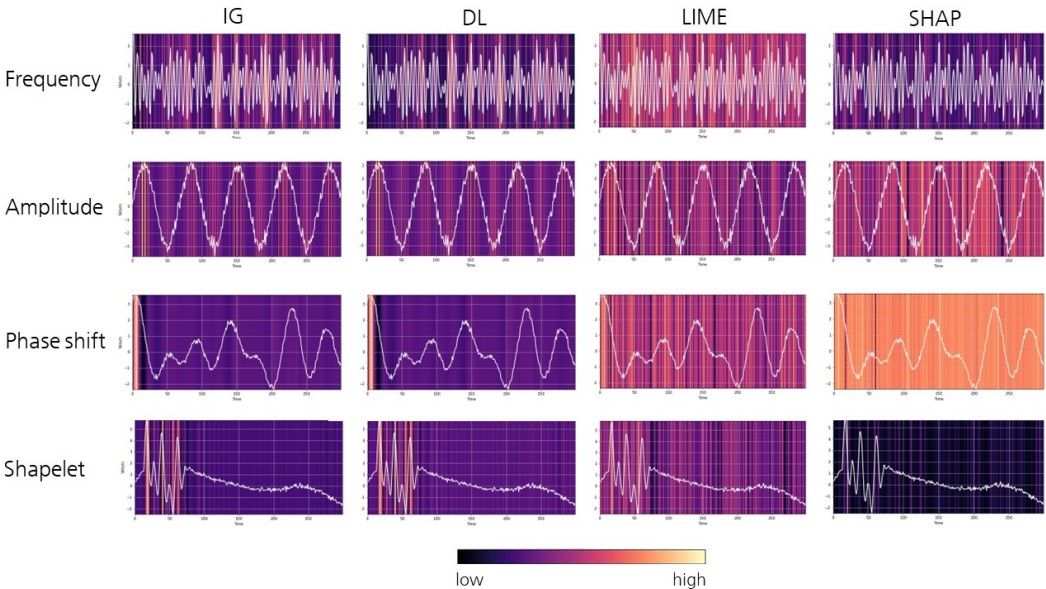

Figure 4: Saliency maps by IG, DL, Lime and SHAP for the CNN+SGT on a frequency, amplitude, phase shift and shapelet experiment, respectively. Explanations by IG and DL clearly focus on aspects related to the latent feature (peaks and valleys for amplitude and frequency, beginning of time sequence for phase shift) and the shapelet. Maps of Lime and SHAP are visually uninterpretable.

Next, to investigate the explainability of the saliency methods, we visualize their output via color-coded heat maps and overlay them onto the original time series (Figure 4). This allows us to assess

the relevance of the saliency scores and the positional information directly. In the shapelet experiment, we expect the maps to highlight the shapelet itself. In the amplitude and frequency experiments, we expect an oscillating heat map with a focus on the peaks (or valleys) and extreme values of the time series, respectively. Finally, in the phase shift experiments, we expect an emphasis on the beginning of the time series. Figure 4 compares the saliency maps of the post-hoc saliency methods (IG, DL, LIME, and SHAP) plotted for one sample per experiment group (shapelet- and latent- experiments). Visual explanations provided by IG and DL align with our expectations for all experiments and are comparatively easy to interpret. For example, in the amplitude experiments, IG and DL highlight the peaks whose values are the direct proxies for the latent feature. On the other hand, the heat maps of SHAP and LIME do not yield the expected visual patterns.

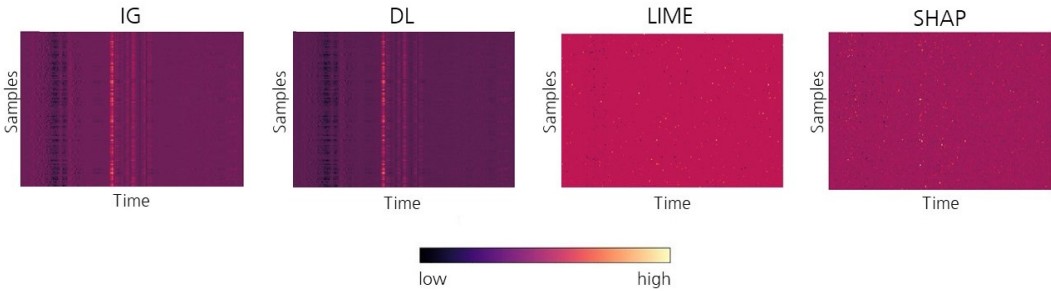

Figure 5: Saliency heat map of IG, DL, LIME and SHAP across all samples of the positive class (occurrence of a shapelet) in the test data set. The IG and DL heat maps show a clear saliency pattern in the middle of the time series, in which the shapelet occurred. The SHAP and LIME heat maps, however, resemble a random saliency assignment.

We expected that the four saliency methods perform reasonably, at least for the shapelet experiments. To investigate this further on the entire data set, we generated Figure 5. These heat maps depict aggregated scores of the saliency methods for the middle-positioned shapelet experiment. In these maps, each row represents a test sample and each column a time point. Figure 5 shows that both SHAP and LIME fail to discover the shapelet pattern across the entire data set. The other two methods IG and DL, however, performed successfully: their aggregated heat map clearly highlights the position of the middle shapelet.

## 5 DISCUSSION

***Promised effectiveness of saliency methods for shapelet-related classification*** The goal of this paper was to demonstrate the fundamental problem of adopted saliency methods for time series data in latent-related classification problems. The methods were expected to be effective in case of the presence of positional information, i.e., shapelets. However, experiments show that some of the methods performed poorly, even in simple shapelet scenarios. In particular, explanations provided by different methods mostly did not align. This finding is in accordance with Neely et al. (2021). Our observation raises caution regarding the use of saliency methods for time series data, previously pointed out by Loeffler et al. (2022); Parvatharaju et al. (2021); Schlegel & Keim (2021). In our findings, IG and DL showed reliable performances throughout the experiments when paired with effective classifiers. Nevertheless, we encourage using various explainability methods as multiple explanations can coexist Wiegreffe & Pinter (2019).

***Need for latent feature saliency methods for time series classification*** We emphasize the need for developing latent feature saliency methods for time series classification. Adopted image saliency methods cannot parse explainable and meaningful saliency scores for time series data with class-distinctive latent patterns. As discussed in Section 2.2, we proposed a definition for "promising failing" methods as ones that produce positional scores associated with informative latent parameters. In the case of Fourier series models, this corresponds to highlighting peaks or valleys, highest peaks, or early time points in case of frequency-, amplitude- and phase-shift-related classification problems, respectively. Not all SoA methods could exhibit such behavior. We hypothesize that this was caused by the independence assumption between neighboring data points, which is made by

the tested approaches. Under this assumption, the model neglects the relative temporal ordering of input features, leading to the inability to detect temporal dependencies. This finding is also reported by Lim et al. (2021).

We observed that the IG and DL methods consistently performed well for shapelet-related problems and produced useful saliency maps for latent-related problems. Note that despite calling these methods "promising", the need for directly scoring the latent parameters remains. We expect this problem to exacerbate for latent-related settings whose features contain less legible associations with the positional information, e.g., rates of changes in state-space models.

***Future work*** To extend the empirical investigations, we suggest considering other time series latent models. We further encourage the development of methods that can incorporate multiple feature spaces into the saliency analysis. With this regard, there is a potential for extracting latent saliency scores directly from positional saliency maps, given that the target latent model is known. Our findings show that the output of IG and DL are associated with the Fourier latent model. This approach (i.e., mapping positional scores to latent scores) serves well as a baseline method.

Throughout our study, the evaluation of saliency maps was performed by visual inspection only since the primary purpose of this paper was to formulate the latent feature saliency problem and motivate further investigation of this topic through a simple experimental framework. For future work, we encourage using quantitative evaluation metrics to objectively assess the performance of different saliency methods. Furthermore, we motivate the extension of our experiments to more complex real-world data sets.

Our analyses were done on the sample level, i.e., we studied individual saliency maps to infer the underlying classification mechanism. Intra-class studies of variability and variance of saliency maps might uncover further information regarding the classification.

# 6 CONCLUSION

Explainability of time series models is an uprising field of research. Interpretation and explanation of black-box classifiers are crucial to building trust in AI. Various image saliency methods have been introduced to time series problems. They focus on positional information of the input features, providing spatial explanations. In time series data, however, the class label may depend on a latent model instead of positional information. To the best of our knowledge, the performance and behavior of saliency methods in such settings have not been explored, and neither has a saliency model accounting for latent features been developed. We demonstrated this problem by empirically showing that if the class label is associated with latent features of the time series instead of the presence of a specific shape, common saliency methods do not provide accurate or interpretable explanations. Finally, we presented an outline for future research to develop extensions for existing saliency methods providing latent saliency results based on time-step-wise importance scores. Our work highlights the need for research on latent saliency detection for deep time series classification.

## AUTHOR CONTRIBUTIONS

Conceptualization, M.S., A.Z. and N.A.; methodology, M.S.; software, M.S. and A.Z.; Formal analysis, A.Z.; validation, A.Z.; investigation, M.S.; resources, N.A.; writing—original draft preparation, M.S. and A.Z.; writing—review and editing, N.A.; visualization, M.S.; supervision, N.A.; project administration, N.A.; funding acquisition, N.A. All authors have read and agreed to the published version of the manuscript.

## ACKNOWLEDGMENTS

We thank Oleksandr Zadorozhnyi for his valuable support throughout the course of the research project. We thank Ruijie Chen, Elisabeth Pachl and Adrian Schwaiger for proofreading the manuscript and providing instructive feedback.

FUNDING

This research was funded by the Bavarian Ministry for Economic Affairs, Regional Development and Energy as part of a project to support the thematic development of the Fraunhofer Institute for Cognitive Systems.

CONFLICTS OF INTEREST

The authors declare no conflict of interest. The funders had no role in the design of the study; in the collection, analyses, or interpretation of data; in the writing of the manuscript; or in the decision to publish the results.

REPRODUCIBILITY STATEMENT

The synthetic data generation algorithm is described in Appendix A.1. The specific data sets employed are stated in terms of the sampling intervals of the latent features in Appendix A.2. Implementation details such as employed libraries were provided in Section 3.2. A GitHub repository containing the complete code base can be found at `https://github.com/m-schroder/TSExplainability`.

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

# A APPENDIX

## A.1 SYNTHETIC DATA GENERATION

Based on the Fourier series latent model, a time series $x_t, t = 1, ..., T$ is modeled as

$$
\begin{aligned}
x_t &= a_0 + \sum_{n=1}^{\infty} a_n \cos(\omega_n t) + \sum_{n=1}^{\infty} b_n \sin(\omega_n t) \\
&= a_0 + \sum_{n=1}^{\infty} A_n \cos(\omega_n t + \phi_n) \\
&= a_0 + \sum_{n=1}^{\infty} A_n \sin(\omega_n t + \phi_n + \frac{\pi}{2}).
\end{aligned}
$$

To simulated data, let $\tilde{n}$ represent the number of amplitudes present in the series, i.e. $\forall i > \tilde{n}, A_i = 0$. For simplicity, we consider centered stationary periodic time series in the data generation process, i.e. $a_0 = 0$. In this case, the value at every time step $t$ is calculated as

$$
x_t = \sum_{i=1}^{\tilde{n}} A_i \sin(\omega_i t + \phi_i + \frac{\pi}{2}). \tag{1}
$$

We refer to the notions amplitude $A$, frequency $\omega$, phase shift $\phi$ as *concepts*. The separate Fourier coefficients $A_i, \omega_i, \phi_i$ for $i = 1, ..., \tilde{T}$ are referred to as latent features. The latent features frequency $\omega_i$ and phase shift $\phi_i$ are each sampled from a uniform distribution. The sampling intervals are chosen with respect to the specific intention in the experiment design. To simulate the amplitude parameters $A_i$, a *dominant amplitude* $A_1$ is sampled. The next amplitudes are calculated considering an exponential decay with a fixed rate $dec$:

$$
A_i = A_1 \exp(-i \cdot dec), \quad i = 1, ..., \tilde{n}.
$$

This makes the first frequency i.e. $\omega_1$ to be the dominant frequency of the Fourier series. Throughout the experiments, all time series were generated with an equal length of 300 time steps. i.e. $T = 300$.

For assigning class labels to the time series samples, we consider the following two scenarios.

*Scenario 1: Label based on the presence of a shapelet*
For assigning shape-based labels to the time series, a shapelet is inserted at a random or fixed position into all time series $X \in D$ belonging to one class. The shapelet is a second simulated Fourier series of length $l \leq T$, where $l = $ window-ratio $\cdot T$ for a chosen window ratio. We define the sampling intervals for the latent features of the shapelet to be non-intersecting with the sampling intervals of the latent features of the original time series $X$. The resulting shapelet replaces the original time series in the interval $[j, j + l]$, where

$$
j \sim \mathcal{U}(1, T - l).
$$

*Scenario 2: Label based on differences in the latent features*

Following the investigation of the effectiveness of explainability methods for latent features, we introduce a second simulation scenario where the labels depend on a difference in the sampling distribution of latent features of the time series. This scenario highlights the main focus of this project and represents our novel view of explainability methods for time series. Similar to the first scenario, the time series are sampled as discretized Fourier series with latent variables $\omega, A$ and $\phi$. The latent dependency is induced as follows:

1. Two normal distributions with different means (based on Table 3) are selected for classes 0 and 1. For positive parameters, the distributions are log-normal.

2. Per each class, $N/2$ Fourier parameters are sampled from the given distributions.

3. The rest of the parameters are sampled from the same distribution for both classes.

4. Sampled parameters are given to the deterministic Fourier series in Equation 1 to generate the temporal samples. Rows are then labeled with the associated class, from the corresponding distribution of which the informative parameters are sampled.

## A.2    DATA SET DESCRIPTION

Based on the data generation method described above, we design ten different mechanisms for binary classification of univariate time series. Table 2 lists the parameters and algorithms for assigning labels to each sample. In table 3 the parameters used for sampling the Fourier series are presented. The complete simulation code base can be found in the GitHub repository at `https://github.com/m-schroder/TSXplainability`.

Table 2: Label-making features per experiment. The overlapping ranges refer to the sampling intervals for frequency and phase shift.

| Experiment | Label feature | Description of shapelet |
|---|---|---|
| 1 | Shapelet | Random position, window length of $0.2 *$ sequence length |
| 2 | Shapelet | Fixed position, last $0.2 *$ sequence length time steps |
| 3 | Shapelet | Fixed position, starting at time step $0.4 *$ sequence length with window length $0.2 *$ sequence length |
| 4 | Shapelet | Fixed position, first $0.2 *$ sequence length time steps |
| 5 | Frequency | Overlapping frequency ranges |
| 6 | Frequency | Overlapping frequency ranges |
| 7 | Phase shift | Non-overlapping phase shift ranges |
| 8 | Phase shift | Non-overlapping phase shift ranges |
| 9 | Amplitude | Different dominant amplitude |
| 10 | Amplitude | Different dominant amplitude |

| Exp. | Number of sines | Freq. low | Freq. high | Phase low | Phase high | Dominant amplitude | Decay rate | Noise ratio |
|---|---|---|---|---|---|---|---|---|
| 1 | 10 | $\frac{\pi}{300}$ | $\frac{\pi}{60}$ | $\frac{-\pi}{4}$ | $\frac{\pi}{4}$ | 1 | 0.3 | 0.1 |
| 2 | 10 | $\frac{\pi}{300}$ | $\frac{\pi}{20}$ | $\frac{-\pi}{4}$ | $\frac{\pi}{4}$ | 1 | 0.3 | 0.1 |
| 3 | 10 | $\frac{\pi}{300}$ | $\frac{\pi}{20}$ | $\frac{-\pi}{4}$ | $\frac{\pi}{4}$ | 1 | 0.3 | 0.1 |
| 4 | 10 | $\frac{\pi}{300}$ | $\frac{\pi}{20}$ | $\frac{-\pi}{4}$ | $\frac{\pi}{4}$ | 1 | 0.3 | 0.1 |
| 5 | 10/10 | $\frac{\pi}{300}/\frac{\pi}{100}$ | $\frac{\pi}{20}/\frac{\pi}{2}$ | $\frac{-\pi}{4}/\frac{-\pi}{4}$ | $\frac{\pi}{4}/\frac{\pi}{4}$ | 1 / 1 | 0.3 / 0.3 | 0.1 / 0.1 |
| 6 | 1/1 | $\frac{\pi}{300}/\frac{\pi}{100}$ | $\frac{\pi}{20}/\frac{\pi}{2}$ | $\frac{-\pi}{4}/\frac{-\pi}{4}$ | $\frac{\pi}{4}/\frac{\pi}{4}$ | 1 / 1 | 0.3 / 0.3 | 0.1 / 0.1 |
| 7 | 1/1 | $\frac{\pi}{300}/\frac{\pi}{300}$ | $\frac{\pi}{20}/\frac{\pi}{20}$ | $0/\frac{-\pi}{4}$ | $\frac{\pi}{4}/\frac{\pi}{2}$ | 1 / 1 | 0.3 / 0.3 | 0.1 / 0.1 |
| 8 | 10/10 | $\frac{\pi}{300}/\frac{\pi}{300}$ | $\frac{\pi}{20}/\frac{\pi}{20}$ | $0/\frac{-\pi}{4}$ | $\frac{\pi}{4}/\frac{\pi}{2}$ | 1 / 1 | 0.3 / 0.3 | 0.1 / 0.1 |
| 9 | 10/10 | $\frac{\pi}{300}/\frac{\pi}{300}$ | $\frac{\pi}{20}/\frac{\pi}{20}$ | $0/\frac{-\pi}{4}$ | $\frac{\pi}{4}/\frac{\pi}{4}$ | 1 / 3 | 0.3 / 0.3 | 0.1 / 0.1 |
| 10 | 1/1 | $\frac{\pi}{300}/\frac{\pi}{300}$ | $\frac{\pi}{20}/\frac{\pi}{20}$ | $\frac{-\pi}{4}/\frac{-\pi}{4}$ | $\frac{\pi}{4}/\frac{\pi}{4}$ | 1 / 3 | 0.3 / 0.3 | 0.1 / 0.1 |

Table 3: Overview of simulation parameters of the Fourier series. If two entries are present in one cell, each the classes were sampled from different distributions. The first entry in each cell corresponds to the sampling parameter of class 0, the second entry to class 1.

