# OpenReview forum: "Post-hoc Saliency Methods Fail to Capture Latent Feature Importance in Time Series Data"
_ICLR.cc/2023/Workshop/TML4H — ICLR 2023 Workshop TML4H Poster_

### Official Review · Reviewer_aTeX · 2023-02-27
**Good motivation but limited novelty**

**Rating:** 4
**Confidence:** 3

**Review:**

This paper has a good startpoint that post-hoc saliency methods fail on time series data, and provide experimantal proves for this point.
The writing quality is good.

My concerns are summarized as follows:
1. The motivation of this paper is interesting. However, as a technical paper, I think the authors should provide a solution instead of just throwing out a problem; as a survey paper, I think the experiment is not comprehensive enough. Therefore, I think the novelty is limited.
2. It seems that the validation data is synthetic, what about the performance on some realistic data?

Overall, I think this paper has a good motivation, but only validating several existing methods on time series data is not sufficient. If the authors can propose a new method to tackle this issue, it will become a good paper.

---

### Official Review · Reviewer_fkCb · 2023-02-28
**Review for Paper28**

**Rating:** 7
**Confidence:** 3

**Review:**

To frame and analyze the problem of latent feature saliency detection, this paper assesses the explainability quality of multiple state-of-the-art saliency methods (Integrated Gradients, DeepLift, Kernel SHAP, Lime) on top of various classification methods (LSTM, CNN, LSTM
and CNN trained via saliency guided training) using simulated time series data with underlying temporal or latent space patterns. As a result, Integrated Gradients and DeepLift are identified as potential candidates for latent saliency scores. From my perspective, I think this paper is an important step toward applying saliency methods, which have been widely adopted in the vision field, to the time series domain.

---

### Meta-Review · Area_Chair_LmRV · 2023-03-05

**Recommendation:** Accept (Poster)
**Confidence:** 4

**Metareview:**

The paper assesses the explainability quality of several state-of-the-art saliency methods on top of various classification methods. The reviewers generally appreciate the paper as a good startpoint to apply saliency methods on time series domain. However, the reviewers have a few concerns, including insufficient method validation. I encourage the authors to consider incorporating constructive feedback from the reviewer, ie., add results on some realistic data to improve this work.